

# Residence Time of Energy in the Atmosphere

Osácar Carlos[1], Membrado Manuel[1], and Fernández-Pacheco Amalio[2]

[1]Facultad de Ciencias. Universidad de Zaragoza. 50009 Zaragoza (Spain)
[2]Facultad de Ciencias and BIFI. Universidad de Zaragoza. 50009 Zaragoza (Spain)

**Correspondence:** C. Osácar (cosacar@unizar.es)

**Abstract.** In atmospheric chemistry, a parameter called residence time is defined for each gas as $T = M/F$, where $M$ represents the mass of the gas in the atmosphere and $F$ is the total average influx or outflux, which in time averages are equal. In this letter we extend this concept from matter to energy which is also a conservative quantity and estimate the average residence time of energy in the atmosphere which amounts to about 56 days. A similar estimation for the residence time of energy in the Sun is of the order of $10^7 \mathrm{yr}$, which agrees with the Kelvin-Helmholtz time scale.

## 1 Introduction

When the inflow, $F$, of any substance into a box is equal to the outflow, then the amount of that substance in the box,$M$, is constant. This constitutes an equilibrium or steady state. Then the ratio of the stock in the box to the flow rate (in or out) is called residence time and is a time scale for the transport of the substance in the box

$$t = \frac{M}{F}. \tag{1}$$

We are referring to a substance measurable and conserved. A good example of this type is the parameter defined in atmospheric chemistry as the average residence time of each individual gas, defined as Eq. (1). $M$ is the total average mass of that gas in the atmosphere and $F$ the total average influx or outflux, which in time averages for the whole atmosphere are equal. See, for example (Hobbs, 2000).

In this letter we want to extend the substance that flows from matter to energy, and estimate the average residence time of energy in the atmosphere. At the end of this letter we will briefly analyze this concept for the Sun. Obviously in Eq. (1), $M$ and $F$ will now represent the total amount of energy in these two systems and the energy flux -in or out- respectively.

Both cases correspond to steady state problems because the storage of energy in the Earth´s atmosphere and in the Sun are not systematically increasing or decreasing.

In Section 2, we consider the Earth´s atmosphere as a big box and using the appropriate energy data in Eq. (1) we compute the time of residence.

In Section 3, we estimate the residence time of energy in the Sun and note that this residence time agrees with the Kelvin-Helmholtz (K-H) time scale. In Section 4 we present a brief discussion.





## 2   Forms of energy in the atmosphere and time of residence.

In this section we will use the energy data provided by Peixoto and Oort (1992). The most important forms of energy in the

atmosphere are: the thermodynamic internal energy, U, the potential energy due to Earth´s gravity, $P$, the kinetic energy, $K$, and the latent energy, $L$, related to the phase transitions of water. The values quoted by these authors of energy per unit surface in units of $10^7 \, \mathrm{J\,m^{-2}}$ are:

$$U = 180.3, \quad P = 69.3, \quad K = 0.123, \quad L = 6.38, \quad E = 256.1 \tag{2}$$

For our purpose of computing the time of residence using Eq.(1), now we only need the inputs, and outputs, of energy in the atmosphere. For this aim, we will use the data cited by Schneider (1995). It is common to express them in units of $e = $

$3.45 \mathrm{W\,m^{-2}}$, where $e$ is a percentage of the Solar Irradiance out of the atmosphere. The atmosphere absorbs $25\,e$ of solar energy, $29\,e$ are absorbed at the surface as sensible and latent heats, and finally it absorbs $100\,e$ as long-wave radiation emitted by surface (the radiation emitted by surface is $104\,e$, but $4\,e$ escapes to the space using the called atmospheric window). Thus the total energy input in the atmosphere is:

$$F_i = 25e + 29e + 100e = 154e = 531 \, \mathrm{W\,m^{-2}}. \tag{3}$$

Regarding to the emitted energy flux, we identify two terms: the component emitted towards the space, $66\,e$, and that emitted

to the surface, commonly denoted by greenhouse effect, $88\,e$.

The sum of the outgoing terms coincides with that of the ingoing terms, $F_o = F_i = F$. Thus, using these values of $E$ and $F$, the estimation of the residence time of energy in our atmosphere, $t_a$ is:

$$t_a = \frac{E}{F} = \frac{256 \times 10^7 \, \mathrm{J\,m^{-2}}}{531 \mathrm{W\,m^{-2}}} = 4.82 \times 10^6 \, \mathrm{s} \approx 56 \, \mathrm{days}. \tag{4}$$

## 3   Estimation of the solar energy and the time of residence of energy in the Sun.

In stars like the Sun, the total energy, $E$, is the sum of the gravitational energy, $E_g$, and the thermal energy, $E_t$,

$$E = E_g + E_t. \tag{5}$$

And the Virial theorem (Kippenhan and Weigert, 1994) links these two energy reservoirs:

$$-E_g = 2E_t. \tag{6}$$

Therefore

$$E = \frac{1}{2} E_g. \tag{7}$$

But the Sun's gravitational energy can be easily estimated

$$E_g \approx -\frac{GM_\odot^2}{2R_\odot} = -1.89 \times 10^{41} \, \mathrm{J}. \tag{8}$$





Inserting (8) into (7) we obtain

$$\|E\| = 9.5 \times 10^{40}\,\text{J}. \tag{9}$$

The ratio between $\|E\|$ and the solar luminosity, $L$ ($3.9 \times 10^{26}\,\text{W}$), which constitutes the energy outgoing flux, is our estimation

for the energy residence time in the Sun, $t_\odot$

$$t_\odot = \frac{\|E\|}{L} \approx 2.6 \times 10^{14}\,\text{s} \approx 0.83 \times 10^{7}\,\text{yr}. \tag{10}$$

The Kelvin-Helmholtz (K-H) time scale, see for example (Kippenhan and Weigert, 1994), for the Sun is:

$$t_{KH} \approx \frac{GM_\odot^2}{R_\odot L}\,\text{yr}. \tag{11}$$

This time scale roughly predicts the time needed by the star to settle to equilibrium after a global thermal perturbation. As the K-H time scale is nothing but

$$t_{KH} \approx \frac{\|E_g\|}{L}, \tag{12}$$

the time of residence of energy computed for the Sun in Eq. (10) is basically the same thing that the old K-H time scale. Stix

in (Stix, 2003), for the time scale of energy transport in the Sun, also estimated $3 \times 10^{7}\,\text{yr}$ as the correct order of magnitude.

## 4   Final discussion

In this letter, we have considered our atmosphere as a big box where energy is in equilibrium, and have estimated its residence time. It amounts to about 56 days. When the same idea is applied to the Sun, we obtain $t \approx 0.87 \times 10^{7}\,\text{yr}$ .

In astrophysics, the question: "how long a photon might take to get from the core of the Sun to the surface" has been

frequently put forward. The answer of several authors was $\approx 10^{4}\,\text{yr}$, see (Shu, 1982), (Bahcall, 1989), etc. In 1989, Mitalas and Sills (1992) pointed out that the average step length assumed for a photon diffusing through the Sun by the previous authors was too long. Correcting this step length, they obtained $1.7\,10^{5}\,\text{yr}$. Finally, Stix (2003), invoking the large heat capacity of the interior of the star, corrected the previous result up to a time scale of the order of $10^{7}\,\text{yr}$.

Bearing in mind what is said in Section 3, our conclusion for the residence time of energy in Earth´s atmosphere ($t \approx 56$

days) is that it is the equivalent of what the K-H time scale is for the Sun. Therefore, after a global thermal perturbation, the atmosphere would need about a couple of months to come back to equilibrium.

*Data availability.* The data used for the estimation of residence time in the Earth's atmosphere were extracted from (Peixoto and Oort, 1992) and (Schneider, 1995). The data used in the estimation of the residence time in the Sun were obtained from (Zombeck, 1990).

*Author contributions.* Amalio Fernández-Pacheco conceived the idea; Carlos Osácar and Amalio Fernádez-Pacheco wrote the paper; Manuel

Membrado contributed in the solar part of the letter.



*Competing interests.* The authors declare no conflict of interests.





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
