# Peer review of "Residence Time of Energy in the Atmosphere"

_Nonlinear Processes in Geophysics, 2019_

## Referee Comment (RC1) · Anonymous Referee #1 · 25 Oct 2019

This is a note a describing a very simple computation of the residence time of energy in the atmospheres of the earth and sun. Apparently this has not been previously stated in the published literature. The note is well written and so I see no need request extensive revision. I do note a few items that could be attended to that would improve the submission. First, it is not exactly correct to say the the earth's atmosphere is energetically in balance. In the past decade there has been a small increase in atmospheric energy, most likely due to anthropogenic increases in greenhouse gases. Second, it would be more consistent if the energy totals and fluxes for the atmosphere were the result of the same computation and from a single source. For example, Peixoto and Oort (1992) could be the single source. Finally the authors should explain why solar absorption of energy is used in the residence time calculation and terrestrial radiation absorption is omitted in this calculation.

---

## Referee Comment (RC2) · Anonymous Referee #2 · 28 Oct 2019

This review concerns tha manuscript entitled 'Residence time of energy in the atmosphere' by O. Carlos, M. Manuel, and R-P. Amalio

The authors use steady state heuristics to find a time scale for the energy in the earth's atmosphere and in the sun that corresponds to the 'residence time' that may be defined for gas components or chemical species in atmospheric chemistry. For the case of the earth, the claim is that the residence time found based on current energetics is equivalent to the K-H time, which is claimed to be the 'residence time' for solar energetics.

[Please forgive my hybrid LaTeX-text formulas below.]

The heuristics used are commonly employed to establish integrated rates–and therefore sort of average timescales–for quantities, q, that satisfy strict conservation laws:

(1) \partial_t q + div (q vector{v}) = 0.

While it is interesting to tie together the energetics of the earth's atmosphere with that of the sun (or other stars), making the claim that the residence time of the earth atmospheric energy and the solar K-H processes are equivalent isn't particularly original, since 'residence times' in each case derive from equation (1), which you can see by integrating over the relevant (spherical) vollumes.

dQ/dt = \integral q \vector{v} dV = \integral_surface q\vector{v} . dS, = f_q A = F_Q

f_q is the 'flux of q' through any portion of the surface, and F_Q is the net inflow/outflow rate through the surface, A. It does not matter what q is, a sort of 'average' 'q-rate' can be defined by Q/T $\sim$ dQ/dt = F_Q. This may be used to define a timescale T $\sim$ Q/F_Q, so _of course_ the time scale for earth's atmosphere energy residence time is equivalent to the K-H timescale: they both come from the exact same conservation law, and this is the same law that describes conservation of mass in the atmospheric chemistry context. The time scale can always be interpreted as the time required to 'deplete' a value of Q, if its rate-of-depletion is F_Q.

So, I'm just not sure what this article is purporting to do, to be honest, other than to perhaps give an explicit interpretation of geophysical and astrophysical processes in terms (i.e., 'residence time') of concepts familiar in atmospheric chemistry. Am I missing something? Even though it's interesting to see actual numbers for the energetics (especially for the earth), the concepts presented here are not new, and are, in any case, very imprecise. The connection of the earth energy residence time to the the K-H time, which is evidently one of the main conclusions, isn't at all surprising given the starting point. Based on these observations, I must decline the manuscript, but will reconsider if the authors can provide some new ideas or at least some new implications, or even a new way of looking at things.

Below are a few other comments about the manuscript that may be worth considering:

(1) Intro: Text beginning with 'Both cases correspond to steady state problems..': This is true strictly at the current epochs of both solar and earth systems because the flux

rates change over geological and stellar lifetimes.

(2) Text beginning 'but 4e escapes to the space using the called atmospheric window).' should read 'but 4e escapes to space using the so-called {\it atmospheric window}).'

(3) Text that reads 'component emitted toward the space, 66e, and that emitted to the surface, commonly denoted by greenhouse effect, 88e.' should read 'component emitted spaceward, 66e, and that emitted towards the surface, commonly denoted by the {\it greenhouse effect}, 88e.'

(4) Regarding Eq. (10): I agree with this formula for the sun; however the same conservation principe applies to larger stars, like O and B-type stars, which have stellar winds that are 4-5 orders of magnitude greater than our sun, and have 'luminosities' comparable to or greater than their star's radiative luminosity. For these cases, the total luminosity must be replaced with L_p + L_w, where L_w = \dot{M} v_\infinity^2, and \dot{M} is the stellar wind mass loss rate, and v_\infinity is the terminal velocity of its wind, and L_p refers to the 'photonic' luminosity. So, L_w becomes the 'wind luminosity'. But in this case, '||E||" must, as you say, include the total 'free energy', which should be understood to _include_ the kinetic energy of the wind at the photosphere, and the connection to the question 'how long a photon might take to escape to the surface' becomes a bit more nuanced. But the connection to the K-H timescale I think still remains, as you've indicated.

(5) Near line 60 with text beginning with 'Therefore, after a global thermal perturbation...': This is a very coarse rule of thumb that follows only from the simple conservation principle you've used. In reality, the earth's atmosphere is a turbulent fluid, and the time to establish equilibrium after a large scale perturbation will be governed by how quickly the cascade of energy takes place from the scales where the perturbation occurs to the scales where dissipation occurs (given that all other energy inputs you've identified remain the same). This can lead to a equilibration time that is significantly shorter that 56 days because of so-called turbulent mixing, which is not an equilibrium

process

---

## Author Comment (AC1) · 4 Nov 2019

i) Carbon dioxide is the most important anthropogenically produced greenhouse gas, and its significant growth in the atmosphere is primarily due to the combustion of fossil fuels, as coal, oil and natural gas. These combustions on surface constitute an additional energy inflow to the atmosphere not considered in Section 2. Its magnitude can be estimated from public data. In recent years, the consumption of fossil fuels has been about 10 Gtoe per year; this implies an energy flux of 0.08 W/m**2 which is small compared with the natural energy fluxes mentioned in Section 2. The data comes from J. Houghton, "Global Warming. The complete briefing". 4th. edition. Cambridge University Press.

ii)We agree with your point and in the new version of the paper all data will come from the reference of Peixoto and Oort.

iii) In the paper, our purpose for introducing the Sun in the discussion was, in part, to gain perspective for the meaning of the residence time of energy in the atmosphere. The fact that the K-H time scale (10**7 yr) was formally like our eqs. (1) and (10), and the fact that the K-H scale is roughly the time needed by the star to settle to equilibrium after a global thermal perturbation was a well known fact in solar physics. See, for example: H. Spruit, Space Science Reviews, 94, 113-126 (2000). Besides, Stix (2002) carried out a numerical experiment using a stellar evolution code. He produced a perturbation in the centre of the Sun that consisted in an increase of the nuclear energy release. The time taken for the adjustement to a new thermal equilibrium ocurred in roughly 10**7 yr. This confirmed the interpretation of the K-H time scale as the time needed by the star to settle to equilibrium after a global perturbation. Stix(2002) also argued why the photon-diffusion time scale in the Sun, in other words, the time scale of energy transport, was coincident with the K-H scale. Our interpretation of your (iii) point is why we have not carried out a calculation of the residence time of energy using the diffusion of photons in the atmosphere. Our purpose is to consider this calculation in a near future using a simple model for the atmosphere and, perhaps, a Monte Carlo method. Anyway, this does not look a simple enterprise and would constitute the subject of another paper.

---

## Author Comment (AC2) · 6 Nov 2019

In our opinion, the concept of residence time can be introduced without starting with the continuity differential equation.

$$E \ / \ F = \tau \ (1)$$

Let assume that there is a system with two elements: a box and a conserved substance. In (1), E is the stock of substance in the box, and the inflow, Fi, and the outflow, Fo, are equal, Fi=Fo=F; then E and F are constant, the system is in a stationary state and $\tau$ is the residence time of that substance in the box.

As you say, in a second interpretation, $\tau$ can be considered as the time to deplete the box in a value E if the rate of depletion is F. This comment will be added in the new version.

[Figure]

It is worth to remember that the Kelvin-Helmholtz (K-H) time scale, which is a relation of type (1), with a value of $10^{**}7$ yr, was originally proposed as an estimation of the life time of the sun. This second interpretation of (1) – depletion time – was incorrect because $10^{**}7$ yr is not the life time of the Sun. The correct interpretation is the first one i.e. $\tau$ is the residence time of energy in the Sun, and therefore, after a global thermal perturbation to the Sun, the K-H time scale is the scale for a new equilibrium. See, for example, H. Spruit, Space Science Reviews, 94, 113-126 (2000).

We do not consider a demerit that Eq. (1), which is traditionally used in atmospheric chemistry and in many other fields, can be applied for the first time to the energy in the atmosphere. We have written this paper because we think that its content is original and correct.

With respect to the other comments,

1) Yes, we agree. We are using data of actual times. Some sentences in this respect will be included in the new version.

2) and 3) Sincere thanks for your English corrections.

4) Yes, for the type of stars you mention, $||E||$ should be modified.

5) In the interior of the Sun, the material is also a turbulent fluid and in a numerical experiment carried out by Stich (2002; see the complete reference in the paper) using a stellar evolution code, he produced a sudden perturbation in the centre of the Sun. This perturbation consisted in a 2% increase of the cross section of the p+p reaction. The time for the adjustement to a new thermal equilibrium was roughly $10^{**}7$ yr, illustrating the interpretation that the K-H time scale is the time scale for a new equilibrium after a global thermal perturbation.

---

## Author Response (AR1)

**Residence Time of Energy in the Atmosphere**

Osácar Carlos[1], Membrado Manuel[1], and Fernández-Pacheco Amalio[2]

[1]Facultad de Ciencias. Universidad de Zaragoza. 50009 Zaragoza (Spain)
[2]Facultad de Ciencias and BIFI. Universidad de Zaragoza. 50009 Zaragoza (Spain)

**Correspondence:** C. Osácar (cosacar@unizar.es)

**Abstract.** In atmospheric chemistry, a parameter called residence time is defined for each gas as $T = M/F$, where $M$ represents the mass of the gas in the atmosphere and $F$ is the total average influx or outflux, which in time averages are equal. In this letter we extend this concept from matter to energy which is also a conservative quantity and estimate the average residence time of energy in the atmosphere which amounts to about  58 days. A similar estimation for the residence time of energy in the Sun is of the order of $10^7 \mathrm{yr}$, which agrees with the Kelvin-Helmholtz time scale.

*In this version, we have introduced new paragraphs and references and omitted some others that we considered unnecessary. The modifications have been marked according to the norms of the journal. With these modifications we have tried to meet the criticism raised by the two referees, which we sincerely acknowledge. In our opinion, the new version has gained in clarity for the reader.*

*Several English corrections suggested by Referee 2 have been introduced in the new version of the paper.*

**1 Introduction**

*After computing the Residence Time of Energy in the Atmosphere (about two months) we were somewhat doubtful about its interpretation. In this sense, the same computation in the Sun was useful because we found that what we called time of residence of energy in the Sun ($10^7$ yr), solar physicists called it Kelvin-Helmholtz time scale. And this time scale represents the time needed by the Sun to settle in a new equilibrium after a global thermal perturbation (Kippenhan and Weigert 1994, Spruit 2000, Stix 2003).*

*Referee 2 suggested to derive the concept of residence time from the continuity equation of fluid mechanics. In our opinion, this simple concept can be maintained as we have introduced it in the paper, since textbooks introduce it similarly.*

When the inflow, $F$, of any substance into a box is equal to the outflow, then the amount of that substance in the box, $M$, is constant. This constitutes an equilibrium or steady state. Then the ratio of the stock in the box to the flow rate (in or out) is called residence time and is a time scale for the transport of the substance in the box

$$\underline{t}\tau = \frac{M}{F}. \tag{1}$$

We are referring to a substance measurable and conserved. A good example of this type is the parameter defined in atmospheric chemistry as the average residence time of each individual gas, defined as Eq. (1). $M$ is the total average mass of that gas in the atmosphere and $F$ the total average influx or outflux, which in time averages for the whole atmosphere are equal. See, for example (Hobbs, 2000).

*An alternative way of considering Eq. (1), (t=M/F), was suggested by Referee 2. It is related to the first interpretation received by the K-H time scale. The K-H scale was originally proposed as an estimation of the life time of the Sun. This would correspond to interpreting t in Eq. (1) as the time of depletion of an amount of energy M of the box.*

In this letter we want to extend the substance that flows from matter to energy, and estimate the average residence time of energy in the atmosphere. At the end of this letter we will briefly analyze this concept for the Sun. Obviously in Eq. (1), $M$ and $F$ will now represent the total amount of energy in these two systems and the energy flux -in or out- respectively.

Both cases correspond to steady state problems because the storage of energy in the Earth´s atmosphere and in the Sun are not systematically increasing or decreasing.

In Section 2, we consider the Earth´s atmosphere as a big box and using the appropriate energy data in Eq. (1) we compute the time of residence.

In Section 3, we estimate the residence time of energy in the Sun and note that this residence time agrees with the Kelvin-Helmholtz (K-H) time scale. In Section 4 we present a conclusion.

**2 Forms of energy in the atmosphere and time of residence.**

*One of the modifications was suggested by Referee 1. In the first Version, the data of energy in the atmosphere were taken from Peixoto & Oort (ref. 4 in the first version). However, in that reference, the flux of infrared radiation from the surface to the atmosphere IR(S->A), and its inverse IR(A->S), are not explicitly specified. These authors only quote the difference of these two fluxes. For our computation of the Residence Time of Energy in the Atmosphere we need to know the individual value of these IR fluxes. Thus, in the new section 2, we have moved to Hartmann's (1994) as the reference that provides both the data of atmospheric energy and the data of energy inflow and outflow in the atmosphere.*

*Referee 1 also raised the following point: in section 2, why had we omitted the anthropogenic contribution to the rest of fluxes? The reason is that this contribution is very small. Using data cited by Houghton we illustrate this question.*

In this section we will use the energy data provided by Hartmann (1994). The most important forms of energy in the atmosphere are: the thermodynamic internal energy, U, the potential energy due to Earth´s gravity, $P$, the kinetic energy, $K$,  the latent energy, $L$, related to the phase transitions of water and $E$, the total energy. The values quoted by  this

author of energy per unit surface in units of  $10^6\,\mathrm{J\,m^{-2}}$ are:

$$U = 180.3\underline{1800}, \quad P = 69.3\underline{700}, \quad K = 0.123\underline{1.3}, \quad L = 6.38\underline{70}, \quad E = 256.1\underline{2571} \tag{2}$$

55  For our purpose of computing the time of residence using Eq.(1), now we only need the inputs, and outputs, of energy in the atmosphere.  It is common to express them in units of  $\underline{e = 3.42\,\mathrm{W\,m^{-2}}}$, where $e$ is a percentage of the Solar Irradiance out of the atmosphere. The atmosphere absorbs  $\underline{20e}$ of solar energy, $29e$ are absorbed at the surface as sensible and latent heats, and finally it absorbs $100e$ as long-wave radiation emitted by surface (the radiation emitted by surface is  $\underline{110e, \text{but } 10e}$ escapes to the space using the  $\underline{\text{so-called}}$ atmospheric

60  window). Thus the total energy input in the atmosphere is:

$$F_i = \underline{25e}\,20e + 29e + 100e = \underline{154e}\,149e = \underline{531}\,509.6\,\mathrm{W\,m^{-2}}. \tag{3}$$

Regarding to the emitted energy flux, we identify two terms: the component emitted  $\underline{\text{spaceward}, 60e}$, and that emitted to the surface, commonly denoted  $\underline{\text{as the}}$ greenhouse effect,  $\underline{89e}$.

The sum of the outgoing terms coincides with that of the ingoing terms, $F_o = F_i = F$. Thus, using these values of $E$ and $F$, the estimation of the residence time of energy in our atmosphere, $t_a$ is:

$$t_a = \frac{E}{F} = \frac{256 \times 10^7\,\mathrm{J\,m^{-2}}}{531\,\mathrm{W\,m^{-2}}} = 4.82 \times 10^6\,\mathrm{s} \approx 56\,\mathrm{days}\; \underline{\frac{2571 \times 10^6\,\mathrm{J\,m^{-2}}}{509.6\,\mathrm{W\,m^{-2}}} = 5.05 \times 10^6\,\mathrm{s} \approx 58\,\mathrm{days}} \tag{4}$$

65  $\underline{\text{We have not considered any anthropogenic contribution because it is negligible compared with the fluxes of solar origin}}$ $\underline{\text{mentioned above. In recent years, the consumption of fossil fuels has been about 10 Gtoe per year; this implies an energy flux}}$ $\underline{\text{of } 0.08\,\mathrm{W\,m^{-2}} \text{ (Houghton, 2004)}}$

**3  Estimation of the solar energy and the time of residence of energy in the Sun.**

*In the paper we quote that in solar physics, Stix (2003) showed the agreement between the thermal adjustment*

70  *time scale and the photon diffusion time scale, and then Referee 1 asked us to do a similar job in the atmosphere. We reply that we will try to do it shortly, but definitely this would be the matter of other paper. It is worth to notice that Stix's paper (2003) appeared more than 14 years after Bahcall's paper (1989).*

In stars like the Sun, the total energy, $E$, is the sum of the gravitational energy, $E_g$, and the thermal energy, $E_t$,

$$E = E_g + E_t. \tag{5}$$

 $\underline{\text{The Virial theorem (Kippenhan and Weigert, 1994) links}}$

75  $\underline{\text{the}}$ two energy reservoirs:

$$-E_g = 2E_t. \tag{6}$$

Therefore

$$E = \frac{1}{2}E_g. \tag{7}$$

 Using the data from (Zombeck, 1990), the Sun's gravitational energy can be easily estimated

$$E_g \approx -\frac{GM_\odot^2}{2R_\odot} = -1.89 \times 10^{41}\,\text{J}. \tag{8}$$

Inserting (8) into (7) we obtain

$$\|E\| = 9.5 \times 10^{40}\,\text{J}. \tag{9}$$

The ratio between $\|E\|$ and the solar luminosity, $L$ ($3.9 \times 10^{26}$ W), which constitutes the energy outgoing flux, is our estimation for the energy residence time in the Sun, $t_\odot$

$$t_\odot = \frac{\|E\|}{L} \approx 2.6 \times 10^{14}\,\text{s} \approx 0.83 \times 10^7\,\text{yr}. \tag{10}$$

The Kelvin-Helmholtz (K-H) time scale,  for the Sun is:

$$t_{KH} \approx \frac{GM_\odot^2}{R_\odot L}\,\text{yr} = 3 \times 10^7\,\text{yr} \approx \frac{E_g}{L}. \tag{11}$$

which is of the order of magnitude of the residence time of energy in the Sun $t_\odot$.

This time scale roughly predicts the time needed by the star to settle to equilibrium after a global thermal perturbation  (Kippenhan and Weigert, 1994), (Spruit, 2000) and (Stix, 2003).

The K-H

$$t_{KH} \approx \frac{\|E_g\|}{L},$$

 scale was originally, proposed as an estimation of the life time of the Sun. This would correspond to interpreting $\tau$ in Eq. ( 1) as the time of depletion of an amount of energy $M$ of the box.

**4   Conclusion**

*With respect to the comment #5 of Referee 2 about the amount of time needed to return to the equilibrium after a global thermal perturbation, we reply that the time of residence of energy in the Sun is the time needed to settle to equilibrium after a global thermal perturbation; we recall the numerical experiment that Stix reports in his paper (Stix 2003). In a model of the Sun's structure, he increased the proton-proton cross section in a 2% and ran the model until a new equilibrium state was reached. The time scale elapsed was about $10^7$ yr.*

In this letter, we have considered our atmosphere as a big box where energy is in equilibrium, and have estimated its residence time. It amounts to about  58 days. When the same idea is applied to the Sun, we obtain  $t \approx 0.83 \times 10^7$ yr.

100    In astrophysics, the question: "how long a photon might take to get from the core of the Sun to the surface" has been frequently put forward. The answer of several authors was $\approx 10^4$ yr, see (Shu, 1982), (Bahcall, 1989), etc. In 1989,  Mihalas and Sills (1992) pointed out that the average step length assumed for a photon diffusing through the Sun by the previous authors was too long. Correcting this step length, they obtained  $1.7 \times 10^5$ yr. Finally, Stix (2003), invoking the large heat capacity of the interior of the star, corrected the previous result up to a time scale of the order of $10^7$ yr. Thus, this
105    author showed the agreement between the thermal adjustment time scale and the photon diffusion time scale.

Bearing in mind what  has been said, our conclusion for the residence time of energy in Earth´s atmosphere ($t \approx$  $t \approx 58$ days) is that it is the equivalent of what the K-H time scale is for the Sun. Therefore, after a global thermal perturbation, the atmosphere would need about a couple of months to come back to a new equilibrium.

*Data availability.* The data used for the estimation of residence time in the Earth's atmosphere were extracted from (Hartmann, 1994). The
110    data used in the estimation of the residence time in the Sun were obtained from (Zombeck, 1990).

*Author contributions.* Amalio Fernández-Pacheco conceived the idea; Carlos Osácar and Amalio Fernádez-Pacheco wrote the paper; Manuel Membrado contributed in the solar part of the letter.

*Competing interests.* The authors declare no conflict of interests.

**References**

115    Bahcall, J. N.: Neutrino Astrophysics, Cambridge University Press, 1989.

Hartmann, D. L.: Global Physics Climatology, Academic Press, 1994.

Hobbs, P.: Introduction to Atmospheric Chemistry., Cambridge University Press, second edn., 2000.

Houghton, J.: Global Warming. The complete briefing 4th Edition, Cambridge University Press, 2004.

Kippenhan, R. and Weigert, A.: Stellar Structure and Evolution, Springer Verlag, 1994.

120    Mihalas, R. and Sills, K.: On the photon diffusion time scale for the sun, The Astrophysical Journal, 401, 759–761, 1992.

Shu, F. H.: The Physical Universe: An Introduction to the Astronomy, Springer Verlag, 1982.

Spruit, H.: Theory of solar irradiance variations, Space Science Reviews, 94, 113–126, https://doi.org/10.1023/A:1026742519353, 2000.

Stix, M.: On the time scale of the energy transport in the Sun, Solar Physics, pp. 3–6, 2003.

Zombeck, M. V.: Handbook of Space Astronomy and Astrophysics, Cambridge University Press, http://ads.harvard.edu/books/hsaa/, 1990.

---

## Author Response (AR2)

Dear Editor,

We accepted the two suggestions made by the Editor (Zoltan Toth) in his decision of February the 5th. We have uploaded a new version of the manuscript including these changes and the file with the differences.

We, also, are waiting to receive the comments from the second round of reviews, as stated in that Editor Decision.

Kind regards
Carlos Osácar

[revised manuscript text omitted]